# Dengue Vaccination: A Practical Guide for Clinicians

**DOI:** 10.3390/vaccines13020145

**Published:** 2025-01-30

**Authors:** Kay Choong See

**Affiliations:** Division of Respiratory and Critical Care Medicine, Department of Medicine, National University Hospital, Singapore 119228, Singapore; kaychoongsee@nus.edu.sg

**Keywords:** Aedes, antibody-dependent enhancement, flavivirus, serogroup, travel medicine, vector borne diseases

## Abstract

Dengue is a growing global public health challenge, with rising incidence and case fatality rates fueled by urbanization and climate change. The substantial mortality, morbidity, and economic burden associated with the disease underscore the need for effective prevention strategies, including vector control, personal protective measures, and vaccination. This narrative review provides a practical guide for clinicians to ensure the appropriate administration of dengue vaccines to at-risk groups, such as individuals in endemic regions and travelers to these areas. Live-attenuated tetravalent dengue vaccines, including Dengvaxia^®^, Qdenga^®^, and Butantan-DV, have demonstrated efficacy in clinical trials but require careful use due to the risk of antibody-dependent enhancement (ADE). To mitigate this risk, guidelines recommend vaccination primarily for individuals with prior confirmed dengue infection, emphasizing the importance of accessible and affordable point-of-care rapid testing. Co-administration of dengue vaccines with other live-attenuated or inactivated vaccines has been shown to be safe and immunogenic, broadening their potential application. However, live-attenuated vaccines are contraindicated for immunocompromised individuals and pregnant women. Enhancing clinician awareness, expanding diagnostic capabilities, and prioritizing high-risk populations are critical steps to optimize vaccination strategies. Combined with robust prevention programs, these efforts are essential to reducing the global burden of dengue and mitigating its impact.

## 1. Introduction

Dengue is a disease caused by the dengue virus, which comprises four serotypes (DEN-1, DEN-2, DEN-3, and DEN-4) [1,2], exclusively causing acute infections in humans. It is transmitted to humans through the bite of infected *Aedes* mosquitoes. Primary dengue refers to an individual’s first infection with the dengue virus, which can be by any of the four serotypes. As infection with a particular serotype confers lifelong immunity specific to that serotype but provides only short-term cross-reactive immunity to the other three serotypes, repeated dengue virus infection by another serotype can occur. Such cases are termed secondary or heterotypic dengue infection. Unlike the heightened risk of severe disease after a second heterotypic dengue virus infection, the risk decreases with third or fourth infections.

Globally, dengue incidence and case fatality are both increasing [3,4], given conditions such as urbanization and rising global temperatures from climate change that favor proliferation of the *Aedes* mosquito and the spread of dengue [5,6,7]. Few regions are free from dengue. Even in areas which have successfully curtailed local transmission, e.g., with *Wolbachia*-infected mosquitoes, global transmission of dengue means that imported cases persist [8], and outbreaks can still occur [9]. Among nearly 6.5 million cases reported in 2023, the average case fatality rate was 0.11% globally and was up to 0.43% in Africa [10]. Despite the already substantial reported figures, the true burden of dengue is likely significantly underreported and underestimated. According to the World Health Organization (WHO), approximately half of the global population is currently at risk of dengue, with an estimated 100 to 400 million infections occurring annually [11].

The clinical burden of the disease is exacerbated by large outbreaks which could overwhelm the healthcare infrastructure of resource-limited countries [12,13]. Additionally, dengue imposes large direct and indirect costs which can challenge even upper-middle income countries like Colombia [14], Brazil, and Thailand [15]. The total economic cost of dengue has been estimated at USD 8.9 billion annually [16,17], which encompasses loss of human productivity from both non-fatal and fatal cases [18].

Given the huge mortality, morbidity, and financial burden of dengue, prevention of dengue infection would be highly important from both the individual and societal perspectives. Apart from vector control and personal protective measures against mosquito bites, vaccination has been commercially available since 2016 [19]. Nonetheless, healthcare providers are mostly unaware of the key considerations for vaccination, even in dengue-endemic regions of Latin America and the Asia-Pacific [20,21]. Limited knowledge of the dengue vaccine may result in both suboptimal uptake and improper use. This narrative review seeks to outline the key concepts that will assist clinicians in correctly administering the dengue vaccine to the appropriate patients. To incorporate up-to-date information, the PubMed bibliographic database was searched for papers published in 2023 and 2024, using the search term “dengue vaccin*” (“*” is a wildcard operator that acts as a placeholder for any number of characters, allowing for broader search results) and updated on 28 December 2024. Any relevant papers were then added to the author’s own library of articles for this review.

## 2. Virology of Dengue

The dengue virus is an RNA flavivirus comprising four distinct serotypes, each with unique antigens and antigenic variation both within and between serotypes [16]. This enveloped RNA virus encodes three structural proteins (capsid, matrix, and envelope) and seven non-structural proteins (NS1, NS2a, NS2b, NS3, NS4a, NS4b, and NS5) [22]. The envelope glycoprotein (E) is the primary target for neutralizing antibodies and plays a crucial role in receptor binding and membrane fusion. Additionally, the membrane (M) and pre-membrane (pre-M) proteins are located on the viral envelope surface.

Dengue virus is primarily transmitted to humans through the bites of infected female Aedes mosquitoes, particularly *Aedes aegypti*, with *Aedes albopictus* playing a secondary role. In most cases, humans transmit the virus to mosquitoes during symptomatic, pre-symptomatic, or asymptomatic phases, with high viremia increasing the risk of transmission [11]. After the *Aedes* mosquito feeds on an infected individual, the dengue virus undergoes an extrinsic incubation period of 8 to 12 days in the mosquito, influenced by factors such as temperature, virus genotype, and viral concentration. Once infectious, the mosquito can transmit the virus throughout its lifespan. Besides vector-based transmission of dengue, other uncommon transmission routes include maternal transmission to newborns, blood product transfusion, organ transplants, and sexual transmission [11,16].

Upon attachment to human cells, the dengue virus delivers its RNA into the cytoplasm, where it serves as a template for synthesizing viral proteins. These proteins are initially produced as a single polyprotein, which is subsequently cleaved into three structural proteins and seven non-structural proteins [22]. These viral proteins then assemble into new virions that are eventually released to infect other cells. The immune system plays a key role in combating the infection, with antibodies targeting specific viral proteins such as E, pre-M, and NS1 [23].

Following a primary infection, individuals develop long-lasting protection to that specific serotype (i.e., homotypic immunity), but immunity to other serotypes (i.e., heterotypic immunity) is transient [24,25]. In other words, infection with a single serotype confers lifelong immunity specific to that serotype but provides only short-term cross-reactive immunity to the other three serotypes, leaving patients with primary dengue susceptible to secondary dengue in the long term.

Heterotypic immunity is a double-edged sword. While it provides partial immunity, incomplete neutralization of the dengue virus can result in antibody-dependent enhancement (ADE), leading to a more severe disease during a secondary infection with a different viral serotype compared to a primary viral infection [26] (Figure 1). In this process, non-neutralizing antibodies form complexes with the dengue virus, which paradoxically facilitate viral cell entry (extrinsic ADE) and viral replication (intrinsic ADE) in host monocytes and macrophages [27]. Additionally, infants born to mothers with dengue antibodies are at risk of ADE, particularly between 6 to 9 months of age. During this period, maternal IgG levels transferred trans-placentally decline to sub-neutralizing levels, increasing the relative risk of ADE fourfold compared to 12-month-old infants, by which time maternal IgG would have largely dissipated [28]. Concerns about ADE heavily influence both the understanding of dengue disease and the use of dengue vaccines.

## 3. Clinical Course and Management of Dengue

Symptoms of dengue, including fever, lethargy, bone pain, headaches, abdominal pain, nausea, vomiting, and petechial rash, are nonspecific. In elderly patients, atypical features such as delirium may occur [29]. Endothelial dysfunction increases vascular permeability, causing plasma leakage, which is a defining feature of severe dengue. Severe dengue can also be complicated by circulatory shock (from excessive plasma leakage, bleeding, or myocarditis), multiple organ failure, and occasionally dengue-associated hemophagocytic lymphohistiocytosis [30]. Dengue can lead to organ dysfunction through various mechanisms. Bone marrow suppression and platelet destruction result in leukopenia, thrombocytopenia, and an increased risk of bleeding. Liver involvement is common, with mild to severe elevations in aminotransferase levels, while rare cases may involve direct viral infection of the brain, leading to encephalitis or encephalopathy. Furthermore, as mentioned in the prior section, heterotypic secondary dengue infections heighten the risk of severe dengue due to ADE [31,32,33].

The diagnosis of dengue can be confirmed either indirectly through serology or directly by detecting viral components. Key diagnostic viral components include the dengue non-structural protein 1 (NS1) antigen and viral RNA, the latter identified using polymerase chain reaction (PCR), a form of nucleic acid amplification testing [34]. Serological testing can detect elevated IgM levels, indicating recent infection, while IgG suggests past infection. PCR is the only method to determine the serotype. Laboratory findings like thrombocytopenia and transaminitis are common but nonspecific.

The course of dengue progresses through three phases: febrile, critical, and recovery. In the absence of specific antiviral treatment, the 2009 WHO guidelines recommend supportive care, including adequate hydration, analgesia (avoiding nonsteroidal anti-inflammatory drugs [NSAIDs] such as ibuprofen and naproxen due to the heightened risk of bleeding), rest, and close monitoring for severe signs like hypotension and bleeding [35].

## 4. Efficacy and Safety of Dengue Vaccination in Adults

Dengue vaccine development has made significant progress since Albert Sabin’s pioneering work, in which he propagated and attenuated live dengue virus in mouse brains [36]. Nonetheless, dengue vaccine development has faced significant challenges, including the lack of suitable animal models that accurately replicate the human immune response. While mouse models, such as AG219 mice, may be used for initial in vivo experiments [37], human participants are essential to assess immunogenicity and safety before field testing new vaccines. In the dengue human infection model (DHIM), the dengue viruses used to induce viremia and symptomatic infection are modified to cause only mild illness [38] and have been employed to challenge individuals several months after receiving candidate vaccines or a placebo [39].

The ideal dengue vaccine should provide strong immunity against all four dengue virus serotypes. Vaccine design currently aims at simultaneously generating separate serotype-specific responses across all four dengue serotypes, with three tetravalent live-attenuated vaccines having demonstrated efficacy against dengue infection in Phase III clinical trials. These three vaccines are administered subcutaneously (e.g., in the deltoid or triceps regions of the upper arm [40]), with different numbers of doses required for the full vaccination schedule: Dengvaxia^®^ (Sanofi Pasteur, Lyon, France) needing three doses six months apart [41,42,43,44], Qdenga^®^ (Takeda, Tokyo, Japan) needing two doses three months apart [45,46,47], and the single-dose Butantan-Dengue Vaccine (DV) (Instituto Butantan, São Paulo, Brazil) [48,49,50] (Table 1).

Dengvaxia^®^ consists of four chimeric viruses in which the pre-M and E structural proteins of the yellow fever 17D vaccine have been replaced with those of each dengue virus serotype. The vaccine retains the non-structural proteins of the yellow fever 17D virus, rather than incorporating dengue viral non-structural proteins. Since the majority of CD8 T-cell epitopes for the dengue virus are in its non-structural proteins, Dengvaxia^®^ is not expected to elicit a significant T cell-mediated immune response against dengue, which may mean lower overall efficacy compared to other vaccines containing dengue non-structural proteins, though no head-to-head trials are available.

Existing Dengvaxia^®^ trials, however, demonstrate serotype-specific differences in efficacy, generally pointing towards greater immunogenicity and efficacy against DEN-3 and DEN-4 dengue serotypes than DEN-1 and DEN-2. In a trial among 10,275 healthy children aged 2–14 years in Asia, overall vaccine efficacy was 56.5% at 13 months after the third dose of Dengvaxia^®^ [41], which meant that there was a 56.5% decrease in the number of dengue cases among the vaccinated versus the unvaccinated. Vaccine efficacy was about 75% for DEN-3 and DEN-4 but only 50% for DEN-1 and 35% for DEN-2. Similarly, in a trial among 20,869 healthy children aged 9–16 years in Latin America, overall vaccine efficacy was 60.8% at 13 months after the third dose of Dengvaxia^®^, 74.0% for DEN-3, 77.7% for DEN-4, 50.3% for DEN-1, and 42.3% for DEN-2 [42]. This trial also showed that vaccine efficacy for the prevention of severe dengue leading to hospitalization was 80.3%, highlighting the potential to reduce clinical burden as well as hospital resource utilization.

Qdenga^®^ differs from Dengvaxia^®^ by using DEN-2 rather than the yellow fever virus as the vaccine backbone. In a trial involving 20,071 healthy children and adolescents of 4–16 years of age in Asia and Latin America, overall vaccine efficacy was 80.9% 12 months after the second dose of Qdenga^®^, and vaccine efficacy for the prevention of hospitalization was 95.4% [45]. Overall vaccine efficacy was slightly lower at 74.9% for patients who never had dengue before. As all the non-structural proteins of DEN-2 are included in Qdenga^®^, serotype-specific immunity should favor DEN-2. This notion is supported by this trial, which showed a vaccine efficacy of 97.7% for DEN-2, compared to 73.7% for DEN-1, 62.6% for DEN-3, and 63.2% for DEN-4. Immunity wanes over time, and at 18 months, overall vaccine efficacy was 73.3%, with a larger drop to 66.2% for patients who did not have dengue prior to vaccination [46]. Again, immunity at 18 months was much better maintained against DEN-2 than for the other serotypes, with efficacy rates being 95.1% for DEN-2 compared to 69.8% for DEN-1, 48.9% for DEN-3, and 51.0% for DEN-4.

In contrast to Dengvaxia^®^ and Qdenga^®^, the Butantan-DV vaccine, based on the TV003 formulation, is designed to elicit more balanced neutralizing antibody responses [48]. This is achieved through its composition of four monovalent dengue virus components, each representing a distinct dengue serotype. Each monovalent component contains all the structural and non-structural proteins of the dengue virus, except for DEN-2 non-structural proteins, which have been replaced with DEN-4 non-structural proteins. In a trial involving 16.235 children, adolescents, and adults aged 2–59 years in Brazil, Butantan-DV had a vaccine efficacy of 89.2% among individuals who had dengue prior to vaccination and an efficacy of 79.6% among those who did not have dengue prior to vaccination two years after the single-dose vaccine [49]. However, balanced immunity across the serotypes could not be clinically demonstrated for various reasons. Vaccine efficacy was higher for DEN-1 at 89.5% than for DEN-2 at 69.6%, presumably due to poorer immunogenicity of the DEN-2 component of the TV003 formulation. Vaccine efficacy for DEN-3 and DEN-4 remains unknown as these serotypes were hardly circulating during the trial period in Brazil. As for Qdenga^®^, a longer-term follow-up for Butantan-DV showed declining efficacy, with overall vaccine efficacy dropping to 67.3% at a mean follow-up interval of 3.7 years after vaccination, with efficacy better preserved at 75.8% for DEN-1 compared to 59.7% for DEN-2 [50].

Even though all three vaccines demonstrated short-term safety and mainly injection-related local and systemic reactions [41,42,45,49], the major safety concern over ADE remains [43]. In the context of vaccination, ADE [51] can occur in two ways (Figure 1). Firstly, by heterotypic antibodies that provide partial immunity only. An ideal vaccine would generate highly type-specific immunity without any cross-reactivity and would generate such immunity against every serotype. Such type-specific antibodies can, for instance, target the host-cell receptor binding domain of the dengue virus envelope domain III (EDIII) [52]. However, this option is not currently available for clinical use, and the other strategy would be to generate balanced immunity against all four serotypes, at high enough levels to be neutralizing. A second way that vaccination can lead to ADE is from waning immunity, which would eventually lead to non-neutralizing antibody levels.

As vaccination with a live-attenuated virus is akin to eliciting an immunological response to dengue infection, dengue vaccination in a dengue-naïve individual (seronegative at baseline) is akin to a first infection. Subsequent natural infection after dengue vaccination, if not completely neutralized, is then akin to secondary dengue infection, which could lead to ADE and severe disease. In comparison, vaccination in an individual who had dengue before (seropositive at baseline) is akin to a second or subsequent infection. In this case, live-attenuated vaccines do not induce disease and instead boost immunity against subsequent dengue infections. As one can surmise, current dengue vaccines would benefit seropositive patients but could place seronegative patients at risk for vaccine-related ADE.

Longer-term follow-ups of landmark trials for Dengvaxia^®^ suggest that vaccine-related ADE can occur with serious consequences, particularly in seronegative patients. In a pooled analysis inclusive of Capeding et al. [41] and Villar et al. [42], Hadinegoro et al. [43] showed that children under the age of nine years had a 58% higher risk of hospitalization for severe dengue if they had received dengue vaccination, which ran counter to the overall decreased risk of hospitalization for older children. Using a case–cohort design, Sridhar et al. [44] elucidated that this risk was possibly due to dengue-seronegative status prior to vaccination, and that this risk extended to children older than nine years who were seronegative. While current trials for Qdenga^®^ and Butantan-DV do not demonstrate ADE, it is conceivable that ADE can still occur given the presence of cross-reactive antibodies and waning antibody levels over time. Hence, an extended follow-up of individuals who receive these newer vaccines is necessary to address concerns regarding ADE.

**Table 1 vaccines-13-00145-t001:** Dengue vaccines approved or pending registration for clinical use.

Vaccine	Manufacturer	Vaccine Type	Use and Administration	Author (Year) [Ref]
Dengvaxia^®^ * (Approved)	Sanofi Pasteur (Lyon, France)	Recombinant tetravalent, live-attenuated dengue vaccine, based on a yellow fever virus backbone, without any dengue non-structural proteins. More immunogenic and efficacious against DEN-3 and DEN-4 dengue serotypes than DEN-1 and DEN-2. Elicits cross-reactive antibodies, with non-neutralizing antibodies against DEN-1 and DEN-2. Risk of ADE of dengue infection in dengue-seronegative vaccinees but less risk in seropositive vaccinees	Indicated for secondary prevention in children and adults aged 6–45 years with test-confirmed previous dengue. Three subcutaneous 0.5 mL doses given 6 months apart	Capeding (2014) [41] Villar (2015) [42] Hadinegoro (2015) [43] Sridhar (2018) [44]
Qdenga^®^ (Approved)	Takeda (Tokyo, Japan)	Recombinant tetravalent, live-attenuated dengue vaccine, based on a DEN-2 backbone, with DEN-2 non-structural proteins. Greater immunity and efficacy against DEN-2 dengue serotype than for DEN-1, DEN-3, and DEN-4	Indicated for children and adults aged 4 years and older, regardless of the presence or absence of prior dengue infection. Two subcutaneous 0.5 mL doses given 3 months apart	Biswal (2019) [45] Biswal (2020) [46] Tricou (2020) [47]
Butantan-Dengue Vaccine (DV) ** (Pending registration)	Instituto Butantan (São Paulo, Brazil)	Recombinant tetravalent, live-attenuated chimeric dengue vaccine comprising four monovalent dengue virus components representing each dengue serotype. Each monovalent component contains all the structural and non-structural proteins of the dengue virus (except DEN-2 non-structural proteins). Should generate balanced immunity against all four dengue serotypes, but randomized trial data are available to support efficacy against DEN-1 and DEN-2 only	Efficacious in children and adults aged 2–59 years. Single subcutaneous 0.5 mL dose	Nivarthi (2021) [48] Kallas (2024) [49] Nogueira (2024) [50]

ADE: Antibody-dependent enhancement. * Production of Dengvaxia^®^ halted in early 2024 due to lack of global demand. ** Registration for Butantan-DV is pending. The request for registration was submitted to Anvisa (Brazil’s National Health Surveillance Agency) on 16 December 2024. Butantan-DV is analogous to the TV-003 vaccine developed by the U.S. National Institutes of Health [48].

## 5. Co-Administration, Duration of Vaccine Protection, and Interpretation of Diagnostic Tests Post-Vaccination

Co-administration of dengue vaccines with other live-attenuated and inactivated vaccines has been shown to be safe and immunogenic, with trials demonstrating this for yellow fever, hepatitis A, MMR (measles, mumps, and rubella), DTaP (diphtheria, tetanus, and acellular pertussis), inactivated polio, *Haemophilus influenzae*, and human papillomavirus vaccines. In many dengue-endemic regions, individuals are often exposed to other flaviviruses and may also receive the yellow fever vaccine. Fortunately, Dengvaxia^®^ can be co-administered with the yellow fever vaccine, which may be expected given that the former is based on a yellow fever virus backbone. In an open-labeled randomized trial involving 390 healthy US adults aged 18–45 years, among individuals given three doses of Dengvaxia^®^ (using an accelerated regime of 0, 2, and 6 months rather than a regular regime of 0, 6, and 12 months), the co-administration of the yellow fever vaccine at dose one of Dengvaxia^®^ did not affect antibody responses against either dengue or yellow fever and did not increase the risk of adverse events [53]. Other trials have shown preserved immunogenicity and safety of Dengvaxia^®^ co-administration with a pentavalent combination vaccine containing DTaP, inactivated polio, and *Haemophilus influenzae* in healthy children aged 15–18 months [54], with MMR in children aged 12–15 months [55], and with human papillomavirus vaccine in children aged 9–13 years [56].

Although Qdenga^®^ is based on a DEN-2 backbone and not a yellow fever virus one, it, too, can be co-administered with the yellow fever vaccine. In a randomized trial involving 900 healthy US adults aged 18–60 years, co-administration of the live-attenuated yellow fever vaccine with the first dose of Qdenga^®^ resulted in preserved immunogenicity of both vaccines, without any increased adverse events [57]. In addition, Qdenga^®^ and the inactivated hepatitis A virus vaccine can be co-administered. In a parallel randomized trial involving 900 UK adults aged 18–60 years, the co-administration of hepatitis A vaccine with the first dose of Qdenga^®^ similarly resulted in preserved immunogenicity of both vaccines, without any increased adverse events [58].

The duration of disease prevention afforded by dengue fever vaccines lasts for at least four years, as shown from the long-term follow-up of participants in various trials. Dengvaxia^®^’s serological response and efficacy were maintained at four years after the final, third vaccine dose in healthy participants aged 9–60 years [59]. Qdenga^®^’s serological response and efficacy were maintained at 4.5 years after the final, second dose for healthy children aged 4–16 years [60,61], with a long-term T-cell response at three years after the second dose for healthy children aged 4–16 years [60]. Butantan-DV’s serological response and efficacy were maintained at 3.7 years for healthy adults, adolescents, and children aged 2–59 years [50].

Given the duration of protection against dengue for several years, over the short term, no booster vaccination should be given to healthy individuals. Another reason to avoid booster vaccination in the short term is that there may not be any boosting effect on humoral immunity in individuals who are dengue seropositive at the baseline (before the first vaccine dose). In a randomized trial involving healthy seropositive adults aged 9–50 years in Columbia and the Philippines, the Dengvaxia^®^ booster vaccination 1–2 years after the final, third dose of the primary series did not elicit any clinically significant boosting of neutralizing antibodies across all the serotypes [40]. Nevertheless, as vaccine efficacy wanes [46,50], booster vaccination might be needed at some point and longer-term trials are required to address this.

As vaccines are not fully protective and vaccine efficacy does not reach 100%, dengue infection with any serotype can occur post-vaccination. Diagnosis of such post-vaccination natural dengue infection differs from that in unvaccinated individuals, as vaccines induce the formation of dengue IgM and IgG antibodies. Consequently, after dengue vaccination, serological testing of dengue IgM should not be used to diagnose acute dengue, and testing of IgG should not be used to identify prior dengue infection [62,63]. Rather, dengue NS1 antigen testing and dengue PCR should be used to diagnose acute dengue infection, as these viral components are not generated by vaccination, as shown in separate studies using Dengvaxia^®^ [44] and Qdenga^®^ [63].

## 6. Recommendations for Dengue Vaccination

Given the risk of ADE, dengue vaccination is recommended primarily for individuals with prior dengue infection, in accordance with guideline recommendations (Table 2) and local regulatory authorizations. This approach applies to both residents of dengue-endemic regions and travelers to these areas, aligning with the goal of preventing severe secondary dengue [64]. If used as a travel vaccine, it should be given at least 14 days before travel [65]. Emphasis could be placed on prioritizing vaccination for individuals at a higher risk of severe disease, including older adults and those with comorbidities [29].

In individuals without prior dengue, it is currently harder to recommend dengue vaccination, as this would predispose them to ADE in future, without any clear guidance with regards to re-vaccination to boost immunity. In addition, other personal preventive methods like avoidance of mosquito bites are safer and possibly more convenient than vaccination. To ensure safe vaccination exclusively for those with prior confirmed dengue infection, point-of-care rapid dengue testing [66] should be made available to frontline primary care providers and priced affordably for patients.

Being live-attenuated, current dengue vaccines need to be avoided in immunocompromised individuals and pregnant women. An additional reason immunocompromised individuals may not be suitable for dengue vaccination is that vaccines are generally less immunogenic in immunocompromised patients [67], which may then lead to the generation of non-neutralizing antibody titers and increase the risk of ADE.

Practically, the choice of dengue vaccines may be limited by availability and local authorizations (Table 2). Nonetheless, as more dengue vaccines become available in the future, clinicians may have the opportunity to select vaccines based on efficacy and convenience. From the convenience standpoint, patients are likely to favor vaccines requiring fewer doses. Even within the same vaccine type, a reduction in the number of doses could improve convenience and uptake of vaccination, e.g., 2 rather than 3 doses of Dengvaxia^®^ [68] and 1 rather than 2 doses of Qdenga^®^ [69], though this would necessarily mean some trade-off in terms of efficacy loss.

Nevertheless, a more important consideration for choosing dengue vaccine type could be efficacy against the various serotypes. As no head-to-head trials have been conducted, comparative efficacy among the three vaccines is unknown. What is known is that the various vaccines demonstrate serotype-specific efficacy and immunogenicity, which may then be considered, based on local serotype predominance. Dengvaxia^®^ has higher efficacy for DEN-3 and DEN-4 than for DEN-1 and DEN-2 [70]. Efficacy may additionally depend on the presence or absence of infection prior to vaccination, as demonstrated by baseline dengue serostatus. Qdenga^®^ demonstrates equally high efficacy against all four dengue serotypes in individuals with positive baseline serostatus. However, in those with negative baseline serostatus, its efficacy is higher against DEN-1 and DEN-2 compared to DEN-3 and DEN-4 [61,71,72,73]. It is important to note that genetic variations among serotypes and differences between circulating strains and vaccine strains may impact vaccine efficacy [74,75].

**Table 2 vaccines-13-00145-t002:** Selected authorizations and recommendations for dengue vaccination.

Patient Population	Guideline (Year) [Ref]	Recommendations
Children	World Health Organization (WHO) (2024) [76]	Vaccination with Qdenga^®^ can be given to children aged 6–16 years in regions with high dengue disease burden (seroprevalence of 60% and above [76,77])
Children	U.S. Centers for Disease Control and Prevention (CDC) (2021) [78]	Vaccination with Dengvaxia^®^ can be given to children aged 9–16 years who have laboratory-confirmed evidence of a prior dengue infection and reside in dengue-endemic regions
Children and adults	World Health Organization (WHO) (2024) [76]	In dengue-endemic countries, Qdenga^®^ vaccination may be considered for individuals aged 6 to 60 years with comorbidities if severe dengue outcomes have been documented in this group
Children and adults	European Medicines Agency (2022) [79]	Vaccination with Qdenga^®^ can be given to adults, adolescents, and children aged four years and older. Approval is not restricted by baseline dengue serological status
Child and adult travelers with significant dengue exposure risk	Swiss Society for Tropical and Travel Medicine (2024) [64]	Vaccination with Qdenga^®^ can be given to travelers aged six years and older who have laboratory-confirmed evidence of a prior dengue infection

## 7. Future Directions

There is a critical need for more immunogenic and effective dengue vaccines, particularly those capable of providing a balanced protection against all four dengue serotypes. Such vaccines should elicit homotypic responses to avoid partial immunity and reduce the risk of ADE in subsequent dengue infections, e.g., the live-attenuated TV005 candidate [39], which has 10 times the amount of the DEN-2 component and better DEN-2 immunogenicity than Butantan-DV (TV003) [80]. Nonetheless, as current studies have only demonstrated medium-term protection afforded by dengue vaccines, and as ADE is suggested by vaccination involving Dengvaxia^®^ [40], long-term efficacy and studies are required. In particular, it remains unclear whether the ADE risk is a class effect among all the current live-attenuated vaccines, as levels of antibodies can decline over time to non-neutralizing levels even with the more immunogenic vaccine formulations [80] and since vaccine-generated cytokine-producing T cells and memory B cells have not been definitively shown to provide sufficient a long-term neutralizing capability to avoid ADE [81,82,83].

Existing vaccines may benefit from heterologous vaccination strategies to potentially improve efficacy against all four dengue serotypes, regardless of baseline serostatus. However, prime-boost regimens must undergo rigorous testing for ADE before broader implementation. For example, in a human challenge study, participants who received a tetravalent dengue purified inactivated vaccine followed by a live-attenuated vaccine boost were later inoculated with the DEN-1 strain approximately two years later. Unexpectedly, the vaccination failed to protect against infection and was associated with enhanced inflammation, suggesting the potential for ADE [84].

T cell-mediated cellular immunity can augment the humoral responses against dengue. T cell-priming vaccines could potentially be given as part of a heterologous vaccination regime with other vaccines that primarily generate humoral immunity. In one study, investigators designed a novel multi-epitope antigen to elicit CD8+ T cell responses against the dengue virus E protein across all serotypes, which triggered good T cell responses against all four serotypes in a murine model [85]. This offers an avenue to a vaccine that can provide balanced cross-protection against all dengue serotypes. If this high level of protection could be maintained, then ADE could be avoided. Separately, if vaccination could prime T cells without eliciting humoral responses that may generate sub-neutralizing antibodies, then protection against dengue could occur without risk of ADE. This concept has been demonstrated by a gold nanoparticle-based, multivalent synthetic peptide vaccine, which effectively induced virus-specific CD8+ T cell responses without generating significant anti-dengue virus antibody responses [86].

New vaccine targets, such as the highly conserved dengue virus NS1, could be used for vaccine production. These targets are resistant to escape mutants, less prone to variation by genotype [87], and help avoid ADE [88,89,90,91]. Additionally, a live-attenuated vaccine that uses a single bivalent E glycoprotein immunogen may dually protect against DEN-2 and DEN-4 in macaque models [92]. Artificial intelligence methods have been employed to predict epitopes for vaccine design, and these hold potential for advancing dengue vaccine development [93].

New vaccine platforms other than live-attenuated virus vaccines may be explored to improve efficacy and perhaps provide safe vaccination to immunocompromised and pregnant women, for whom live-attenuated vaccines are currently contraindicated (Figure 2). Virus-like particle (VLP)-based vaccines replicate the virus’s antigenic structure while excluding its viral genome, making them suitable for very young children and immunocompromised individuals. VLPs have been engineered to display the envelope domain III (EDIII) of all four dengue virus serotypes. These vaccines have demonstrated strong immunogenicity, eliciting potent and long-lasting neutralizing antibodies against all four serotypes, in both mice and macaques [52,94]. A similar preclinical study used bacteriophage-derived AP205 VLPs to display EDIII through genetic fusion or chemical coupling. This approach optimized the VLP structure to enhance immunogenicity and induce neutralizing antibody responses against all four dengue serotypes, without producing any enhancing antibodies [95]. In another study, a tetravalent nanoparticle vaccine featuring modified envelope proteins from all four dengue virus serotypes displayed on a ferritin scaffold induced strong humoral and cellular immunity in mice, without causing ADE. It also provided effective protection against lethal DEN-2 and DEN-3 challenges in mice [96]. Separately, a nucleotide-modified mRNA vaccine encoding the membrane and envelope structural proteins of DEN-1, encapsulated in lipid nanoparticles, was tested in mice [97]. It elicited robust serotype-specific antiviral immune responses comparable to those induced by viral infection and provided protection against a lethal dengue virus challenge in immunocompromised mice.

From the public health perspective, dengue vaccination needs to be better studied as it is unclear if mass vaccination can be used to control outbreaks, e.g., Dengvaxia^®^ in Brazil [98]. In the meantime, continued development of non-vaccine methods of vector control, e.g., *Wolbachia*-carrying *Aedes* mosquitoes are needed to complement vaccination for dengue prevention [99,100].

**Figure 2 vaccines-13-00145-f002:**
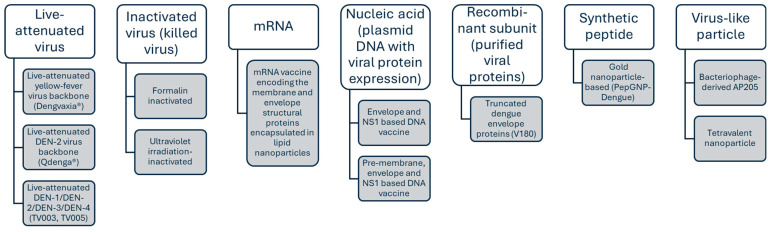
Selected dengue vaccines in clinical use and development. These include vaccines using the following platforms: live-attenuated virus [41,45,48], inactivated virus [101,102], mRNA [97], nucleic acid (DNA) [103,104], recombinant protein subunit [105], synthetic peptide [86], and virus-like particle [95,96].

## 8. Conclusions

In conclusion, dengue remains a significant global public health challenge, with rising incidence and case fatality rates driven by urbanization and climate change. Preventative measures such as vector control, personal protective strategies, and vaccination are critical in addressing the substantial mortality, morbidity, and economic burden associated with the disease. Vaccination requires careful consideration to ensure its safe and effective use. Current live-attenuated tetravalent dengue vaccines, including Dengvaxia^®^, Qdenga^®^, and Butantan-DV, have demonstrated efficacy in clinical trials but are associated with potential risks, particularly ADE. To mitigate these risks, guidelines generally recommend vaccination only for individuals with prior confirmed dengue infection, emphasizing the need for accessible and affordable point-of-care rapid testing.

Continued efforts to enhance vaccine efficacy against all known serotypes are essential to improving protection and mitigating the risk of antibody-dependent enhancement (ADE). Current strategies, including optimized epitope design, heterologous prime-boost approaches, and the use of adjuvants, may be further strengthened by AI-driven epitope prediction. Additionally, robust surveillance systems must monitor emerging mutations and novel strains, such as a potential fifth serotype [106], necessitating adaptive vaccine development. Ensuring the safety of dengue vaccines in vulnerable populations—such as pregnant women and children when these vaccines become available—requires stratified clinical trials that specifically recruit these subgroups, complemented by long-term post-marketing surveillance.

Healthcare providers play a pivotal role in ensuring appropriate vaccine administration, particularly in endemic regions and among high-risk populations such as older adults and individuals with comorbidities. Co-administration of dengue vaccines with other live-attenuated or inactivated vaccines has been shown to be safe and immunogenic, broadening their potential use. However, live-attenuated dengue vaccines remain contraindicated for immunocompromised individuals and pregnant women. Enhancing awareness among clinicians, expanding access to diagnostic tools, and focusing on at-risk populations will be essential for optimizing the impact of dengue vaccination programs, ultimately reducing the global burden of this preventable disease.

## Figures and Tables

**Figure 1 vaccines-13-00145-f001:**
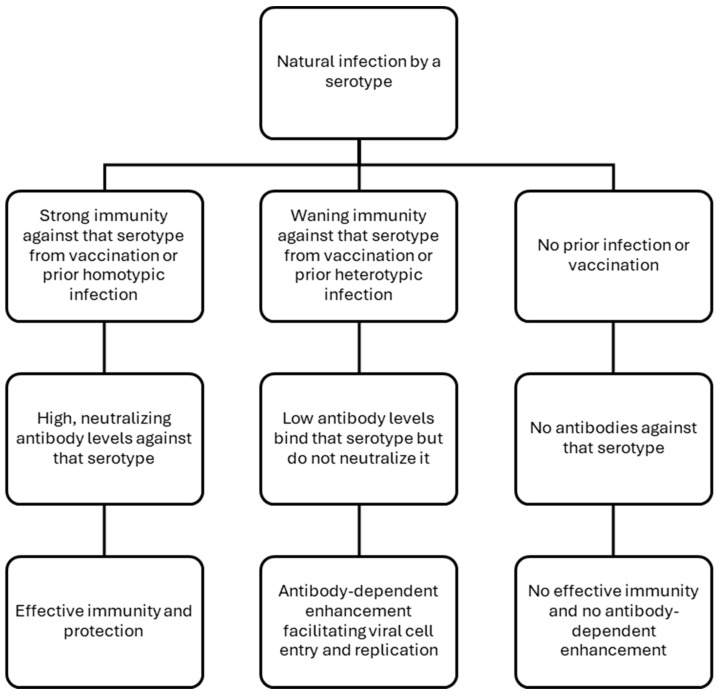
Immune response and antibody-dependent enhancement in dengue infection among individuals aged 12 months or older. For information on dengue infection in infants younger than 12 months, see text.

## Data Availability

All data used can be found in the text and tables.

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
