# Peer review of "Dengue Vaccination: A Practical Guide for Clinicians"

_vaccines, 2025, doi:10.3390/vaccines13020145_

Round 1

Reviewer 1 Report

Comments and Suggestions for Authors

In this manuscript, the author makes a review of dengue and the vaccines that are available up tot date, describing the recommendation to take one of the three options. This is an important theme right now, especially with the number of dengue cases increasing every year. 

The manuscript is relevant and well-written. I recommend only a few edits, listed below. 

Minor Reviews:

Correct the spelling of Columbia. I assume it's the South-American country, which is ColOmbia.

Introduction, line 78: how would fever increase the risk of dengue transmission to the vector?

Virology of dengue section: 2nd, 3rd and 4th paragraphs are missing references. 

I'm not sure if dengue vaccination has been available for over a decade. Could the author explore this sentence more? where is the data about this vaccine? 

Author Response

In this manuscript, the author makes a review of dengue and the vaccines that are available up tot date, describing the recommendation to take one of the three options. This is an important theme right now, especially with the number of dengue cases increasing every year.  The manuscript is relevant and well-written. I recommend only a few edits, listed below.

Minor Reviews:

Correct the spelling of Columbia. I assume it's the South-American country, which is ColOmbia.

[Reply] Thanks for spotting the error. The spelling has been corrected.

Introduction, line 78: how would fever increase the risk of dengue transmission to the vector?

[Reply] Fever itself does not directly increase the risk of dengue transmission but is associated with higher viremia levels. To prevent any misunderstanding, I have removed "fever" from the sentence.

Virology of dengue section: 2nd, 3rd and 4th paragraphs are missing references.

[Reply] References are added to all the paragraphs.

I'm not sure if dengue vaccination has been available for over a decade. Could the author explore this sentence more? where is the data about this vaccine?

[Reply] Clarified that the dengue vaccine has been commercially available since 2016, excluding the years of clinical development.

Reviewer 2 Report

Comments and Suggestions for Authors

The review paper titled "Dengue Vaccination: A Practical Guide for Clinicians" by Kay Choong See offers an insightful discussion of the currently available dengue vaccines and their practical applications. It serves as a valuable resource for clinicians seeking guidance on vaccine usage and an important reference for policymakers.

The paper is well-written, with a primary focus on the available live attenuated dengue vaccine and antibody-dependent enhancement (ADE). Dengvaxia is most associated with ADE among the three currently available vaccines, warranting a clearer explanation. Additionally, significant challenges, such as cross-reactivity with other flaviviruses the existence of multiple DENV serotypes, and virulence, continue to hinder the development of an efficient dengue vaccine.

To enhance the paper's comprehensibility, I recommend including a schematic illustration of ADE mechanisms and the various types of dengue vaccines (e.g., inactivated, DNA, recombinant subunits, peptide-based). A schematic representation of the developmental strategies of the three live-attenuated dengue vaccines would also be beneficial for readers.

I suggest the author expand on challenges in dengue vaccine development, such as the lack of suitable animal models. For instance, the AG129 mice strain, deficient in IFN α/β/γ receptors, is considered highly suitable. What are the authors' perspectives on using this model? Similarly, could the dengue human infection model (DHIM) be a potent tool for vaccine evaluation? The authors might also consider discussing AI in designing vaccines and the potential role of mRNA vaccines in dengue prevention.

How might we overcome challenges to improve vaccine efficacy across all serotypes? Additionally, considering the identification of dengue serotype 5 in Malaysia, how might this impact vaccine strategies? Furthermore, what measures can ensure the safety of current dengue vaccines for vulnerable populations such as pregnant women, children, and both naïve and those previously infected with dengue?

Lastly, I recommend providing a clearer explanation by the author of the WHO's current position on dengue vaccine usage to better guide clinicians. 

Comments on the Quality of English Language

Minor English editing is suggested. 

Author Response

The review paper titled "Dengue Vaccination: A Practical Guide for Clinicians" by Kay Choong See offers an insightful discussion of the currently available dengue vaccines and their practical applications. It serves as a valuable resource for clinicians seeking guidance on vaccine usage and an important reference for policymakers.

The paper is well-written, with a primary focus on the available live attenuated dengue vaccine and antibody-dependent enhancement (ADE). Dengvaxia is most associated with ADE among the three currently available vaccines, warranting a clearer explanation. Additionally, significant challenges, such as cross-reactivity with other flaviviruses the existence of multiple DENV serotypes, and virulence, continue to hinder the development of an efficient dengue vaccine.

To enhance the paper's comprehensibility, I recommend including a schematic illustration of ADE mechanisms and the various types of dengue vaccines (e.g., inactivated, DNA, recombinant subunits, peptide-based). A schematic representation of the developmental strategies of the three live-attenuated dengue vaccines would also be beneficial for readers.

[Reply] Included Figure 1 (Immune response and antibody-dependent enhancement in dengue infection among individuals aged 12 months or older) and Figure 2 (Selected dengue vaccines in clinical use and development).

I suggest the author expand on challenges in dengue vaccine development, such as the lack of suitable animal models. For instance, the AG129 mice strain, deficient in IFN α/β/γ receptors, is considered highly suitable. What are the authors' perspectives on using this model? Similarly, could the dengue human infection model (DHIM) be a potent tool for vaccine evaluation?

[Reply] Included information about the mouse model and the dengue human infection model in the first paragraph of part 4 (Efficacy and safety of dengue vaccination in adults).

The authors might also consider discussing AI in designing vaccines and the potential role of mRNA vaccines in dengue prevention.

[Reply] Included information about AI in the antepenultimate paragraph and information about mRNA vaccines in the penultimate paragraph of part 7 (Future directions)

How might we overcome challenges to improve vaccine efficacy across all serotypes? Additionally, considering the identification of dengue serotype 5 in Malaysia, how might this impact vaccine strategies? Furthermore, what measures can ensure the safety of current dengue vaccines for vulnerable populations such as pregnant women, children, and both naïve and those previously infected with dengue?

[Reply] Included in the conclusion: “Continued efforts to enhance vaccine efficacy against all known serotypes are essential to improving protection and mitigating the risk of antibody-dependent enhancement (ADE). Current strategies, including optimized epitope design, heterologous prime-boost approaches, and the use of adjuvants, may be further strengthened by AI-driven epitope prediction. Additionally, robust surveillance systems must monitor emerging mutations and novel strains, such as a potential fifth serotype, necessitating adaptive vaccine development. Ensuring the safety of dengue vaccines in vulnerable populations – such as pregnant women and children when these vaccines become available – requires stratified clinical trials that specifically recruit these subgroups, complemented by long-term post-marketing surveillance.”

Lastly, I recommend providing a clearer explanation by the author of the WHO's current position on dengue vaccine usage to better guide clinicians.

[Reply] Included more information in Table 2.

Minor English editing is suggested.

[Reply] I used Microsoft Word's built-in Editor to enhance the language in my manuscript.

Reviewer 3 Report

Comments and Suggestions for Authors

The authors reviewed publications up to the end of 2024 concerning “dengue vaccine” to provide a practical guide for clinicians to ensure the appropriate administration of dengue vaccines to at-risk groups. The manuscript is well-written and useful for clinicians to understand the current situation of the dengue vaccines. Specific comments follow.

Major points:

1.     Line 95: Please add a reference for transient heterotypic immunity.

Minor points:

1.     Line 65: dengue vaccin*” should be dengue vaccine”?

Author Response

The authors reviewed publications up to the end of 2024 concerning “dengue vaccine” to provide a practical guide for clinicians to ensure the appropriate administration of dengue vaccines to at-risk groups. The manuscript is well-written and useful for clinicians to understand the current situation of the dengue vaccines. Specific comments follow.

Major points:

  1. Line 95: Please add a reference for transient heterotypic immunity.

[Reply] Reference added.

Minor points:

  1. Line 65: “dengue vaccin*” should be “dengue vaccine”?

[Reply] Clarified the search term “dengue vaccin*,” where “*” is a wildcard operator that serves as a placeholder for any number of characters, enabling broader search results.